# Primary Endoscopic Endonasal Management of Giant Pituitary Adenomas: Outcome and Pitfalls from a Large Prospective Multicenter Experience

**DOI:** 10.3390/cancers13143603

**Published:** 2021-07-18

**Authors:** Salvatore Chibbaro, Francesco Signorelli, Davide Milani, Helene Cebula, Antonino Scibilia, Maria Teresa Bozzi, Raffaella Messina, Ismail Zaed, Julien Todeschi, Irene Ollivier, Charles Henry Mallereau, Guillaume Dannhoff, Antonio Romano, Francesco Cammarota, Franco Servadei, Raoul Pop, Seyyid Baloglu, Giovanni Battista Lasio, Florina Luca, Bernard Goichot, Francois Proust, Mario Ganau

**Affiliations:** 1Neurosurgery Unit, Hautepierre Regional Hospital, Strasbourg University, 67200 Strasbourg, France; schibbaro@hotmail.com (S.C.); helene.cebula@hotmail.fr (H.C.); antonino.scibilia@unibo.it (A.S.); mari-ta@hotmail.it (M.T.B.); ismailzaed1@gmail.com (I.Z.); julien.todeschi@chru-strasbourg.fr (J.T.); irene.ollivier@chru-strasbourg.fr (I.O.); charleshenry.mallerau@chru-strasbourg.fr (C.H.M.); guillaume.dannhoff@chru-strasbourg.fr (G.D.); francesco.cammarota@chru-strasbourg.fr (F.C.); raoul.pop@chru-strasbourg.fr (R.P.); seyyid.baloglu@chru-strasbourg.fr (S.B.); francois.proust@chru-strasbourg.fr (F.P.); mario.ganau@alumni.harvard.edu (M.G.); 2Neurosurgery Unit, Department of Basic Medical Sciences, Neurosciences, Sense Organs, University “Aldo Moro”, 70124 Bari, Italy; raffamessina@gmail.com; 3Neurosurgery Unit, Humanitas Research Hospital, 20089 Milano, Italy; davide.milani@gmail.com (D.M.); franco.servadei@hunimed.eu (F.S.); giovanni.lasio@humanitas.it (G.B.L.); 4Neurosurgery Department, Parma and Reggio Emilia Hospital, University of Parma, 43126 Parma, Italy; antonio.romano@asmn.re.it; 5Endocrinology Unit, Hautepierre Regional Hospital, Strasbourg University, 67200 Strasbourg, France; florina.luca@chru-strasbourg.fr (F.L.); bernard.goichot@chru-strasbourg.fr (B.G.); 6Neurosurgery Department, Oxford University Hospital, Oxford OX3 9DU, UK

**Keywords:** giant pituitary adenomas, pituitary tumors, endoscopy, visual field, visual acuity, pituitary insufficiency, endoscopic endonasal extended approach, trans-tuberculum/transplanum approach, Pituitary Apoplexy

## Abstract

**Simple Summary:**

Giant pituitary adenomas are highly invasive tumors whose treatment is challenging. Surgery is their management mainstay. However, there is no consensus about the type of approach. Open transcranial, microscopic, and endoscopic trans-sphenoidal approaches have all been employed, alone or in combination. Extended endoscopic endonasal techniques may represent a versatile and safe one-stage approach. Our research aimed at evaluating prospectively their applicability, effectiveness, and safety in a multicenter series, to acquire further evidence toward its use in the treatment of those challenging lesions. Ninety-six patients were recruited and followed-up for 52.4 months on average. Most of them (81.2%) presented with visual deficits and >50% had various degrees of adenohypophysis insufficiency. Resection of at least 75% of initial volume was achieved in all cases, with 98.7% visual improvement, >50% endocrine deficit recovery and a permanent complication rate of 4.2%, indicating extended endoscopic endonasal approaches as a valuable treatment option.

**Abstract:**

Purpose: To evaluate factors influencing clinical and radiological outcome of extended endoscopic endonasal transtuberculum/transplanum approach (EEA-TTP) for giant pituitary adenomas (GPAs). Methods: We recruited prospectively all consecutive GPAs patients undergoing EEA-TTP between 2015 and 2019 in 5 neurosurgical centers. Preoperative clinical and radiologic features, visual and hormonal outcomes, extent of resection (EoR), complications and recurrence rates were recorded and analyzed. Results: Of 1169 patients treated for pituitary adenoma, 96 (8.2%) had GPAs. Seventy-eight (81.2%) patients had visual impairment, 12 (12.5%) had headaches, 3 (3.1%) had drowsiness due to hydrocephalus, and 53 (55.2%) had anterior pituitary insufficiency. EoR was gross or near-total in 46 (47.9%) and subtotal in 50 (52.1%) patients. Incomplete resection was associated with lateral suprasellar, intraventricular and/or cavernous sinus extension and with firm/fibrous consistence. At the last follow-up, all but one patient (77, 98.7%) with visual deficits improved. Headache improved in 8 (88.9%) and anterior pituitary function recovered in 27 (50.9%) patients. Recurrence rate was 16.7%, with 32 months mean recurrence-free survival. Conclusions: EEA-TTP is a valid option for GPAs and seems to provide better outcomes, lower rate of complications and higher EoR compared to one- or multi-stage microscopic, non-extended endoscopic transsphenoidal, and transcranial resections.

## 1. Introduction

Pituitary adenomas (PAs) are benign, slow growing tumors that may cause compression and/or encasement of surrounding neural structures, such as optic nerve, chiasm, pituitary stalk, and hypothalamus, and/or vascular structures, such as carotid artery, anterior cerebral artery, anterior communicating artery complex, and cavernous sinus (CS). Clinically, they may present with either signs and symptoms secondary to mass effect (headache, visual dysfunction), or with clinical syndromes related to abnormal hormonal secretion [1,2]. Treatment goal for PAs is two-fold: preservation/re-establishment of adequate pituitary function and decompression of nervous and vascular structures [3,4,5]. In general, management options include medical, surgical, and radiosurgical treatments, such as stereotactic radiotherapy (SRT) and stereotactic radiosurgery (SRS), or a combination of all of them, depending on the clinical status at baseline and the size of the tumor [3,5]. Whenever PAs reach the size of macroadenoma (from 10 up to 40 mm diameter) or larger (thereby classified as ‘‘giant’’, GPAs) surgery is more challenging, due to extension beyond the sella turcica and invasion of adjacent nervous and vascular structures [6]. However, by using minimally invasive endoscopic tools and enlarging the trans-sphenoidal corridor, it is possible to visualize structures beyond the sella, appreciate tumor margins including suprasellar extension and achieve satisfactory quality of resection with acceptable complication rate [7]. In this prospective multicenter study, we enrolled consecutively patients harboring GPAs treated via extended endoscopic endonasal approach transtuberculum/transplanum (EEA-TTP). This study focuses particularly on surgical pitfalls and analyzes the advantages and disadvantages of EEA-TTP as well as the implications of multimodality management for these extremely challenging lesions.

## 2. Materials and Methods

### 2.1. Patient Population

Clinical data were prospectively collected and analyzed from a collaborative multicenter database including all cases of non-secreting GPAs undergoing EEA-TTP resection in 5 different European centers during a 5-year period (from January 2015 to December 2019).

### 2.2. Data Collection

All patients had pre- and postoperative laboratory assessment (total blood count, biochemistry, and ionogram, as well as pituitary hormones serum level) and imaging (CT and MRI scans). Invasion/extension of GPAs within CS was classified according to Knosp grade [8] by a neuroradiologist and confirmed by the multidisciplinary team at each site. All patients included underwent EEA-TTP resection as the only one-staged surgical treatment; histopathological diagnosis was obtained in all patients. From the prospective database, the authors analyzed also preoperative visual and endocrinology status and outcome and surgical complications. Postoperative EoR (extent of resection) was evaluated by 2 independent neuroradiologists and blinded as regards intraoperative findings, on enhanced MRI performed within 48 h after surgery and repeated 3 months postoperatively. Unless contraindicated, all patients were administered thromboembolic prophylaxis starting 48 h after surgery according to guidelines [9,10] and discharged home as soon as safely possible. EoR was classified as follows: gross total resection (GTR) 100% of the initial tumor volume, near total resection (NTR) between 96 and 99% of the initial tumor volume, subtotal resection (STR) between 75 and 95% of the initial tumor volume. Tumor consistency was assessed by at least two surgeons, either intraoperatively or revising surgical videos: it was defined as soft if GPA was resectable with conventional curettage and suction, or firm/fibrous if more resistant to these maneuvers and requiring some degree of mechanical debulking and/or extracapsular dissection [11,12]. Vascularization was also assessed intraoperatively and on video recordings as significant or non-significant depending on degree of intraoperative bleeding appreciated through the endoscopic lens. Long term follow-up implied clinical assessment and MRI at 6 months, 12 months, and yearly afterwards, unless new symptoms occurred.

### 2.3. Endocrinological Evaluation

Baseline tests included insulin-like growth factor 1 (IGF-1), adrenocorticotropic hormone (ACTH), 8 a.m. cortisol, prolactin (PRL), luteinizing hormone (LH), follicle-stimulating hormone (FSH), eostradiol, testosterone, sex hormone binding globulin (SHBG), thyrotropin (TSH), and total thyroxine (TT4).

### 2.4. Ophtalmological Evaluation

The aim of the evaluation was to assess visual acuity, color vision, peripheral vision, eye movements and appearance of retina and optic nerve. It included best-corrected visual acuity (BCVA), slit-lamp biomicroscopy, intraocular pressure, dilated fundus examination, visual field (VF), and optical coherence tomography (OCT) before and 6 weeks, 3, 6, 12, and 24 months after surgery.

### 2.5. Ethics

This study was conducted in accordance with the Principles of Ethics for Medical Research Involving Human Subjects set in the Declaration of Helsinki and its subsequent amendments. To report our results, we followed the recommendations of the STROBE (Strengthening the Reporting of Observational Studies in Epidemiology) statement for observational studies [13].

### 2.6. Statistical analysis

Variables were classified as continuous or categorical. Mean, ranges, and medians were used for continuous data collected in the study. Categorical data were presented as total count and proportions. Extent of resection was analyzed regarding demographic, clinical, radiological, and intraoperative tumor characteristics using the chi-square and Fisher exact tests. A *p*-value < 0.05 was considered statistically significant. Graphpad online calculator was used for all statistical analysis of the study (http://www.graphpad.com/quickcalcs/ accessed on 13 March 2021).

### 2.7. Surgical Technique

Description of EEA-TTP is based on previous technical reports [14,15,16,17,18] and was agreed and shared by the surgeons participating in this prospective study. EEA-TTP shares several technical features with standard endoscopic transphenoidal approach, nonetheless it entails a binostril approach and much wider opening of the upper portion of the anterior wall of sphenoid sinus, which is obtained by removing middle, and if necessary, superior turbinate and realizing a posterior ethmoidectomy. Once the sphenoid sinus was widely opened and all anatomical landmarks were identified, we removed the suprasellar notch and the posterior planum sphenoidale in a postero-to-anterior way, up to a maximum of 20 mm. At this stage, optic nerve protuberances represent the lateral limits of bone removal, creating a trapezoidal shape door to the extradural space. Dura mater was always opened from the midline diverging laterally in a “V” or “Y” fashion over the planum to allow exploration of the supradiaphragmatic compartment. Tumor removal started inferiorly, proceeded laterally toward both sides up to the lateral CS walls, then the suprasellar component was debulked and finally GPA’s capsule was separated, whenever possible, from the surrounding neurovascular structures, using sharp dissection. At the end of procedure the skull base defect was reconstructed in a multilayer fashion by gasket seal reconstruction, covered by a pedicled nasoseptal flap [19]. No peri- or postoperative lumbar drain was used primarily. All procedures were performed by a multidisciplinary team of Neurosurgery/ENT specialists, with the assistance of intraoperative navigation and microdoppler. Excision of each GPA was tailored to the extension of the specific lesion (i.e., tumor invading nasal or paranasal cavities, clivus, encasing vessels, and nerves) by always prioritizing patients’ safety vs. EoR.

## 3. Results

A total of 96 (8.2%) GPAs were treated by EEA-TTP and included in this prospective collaborative multicenter study among 1169 patients harboring PAs and endoscopically managed during a 5-year period. All centers contributed equally to patients’ recruitment. Demographics, patients’ clinical and radiological features, and outcome are summarized in Table 1. 

Among the 96 patients included, 55 were men (57.3%) and 41 were women (42.7%), and the mean age was 52.2 years (range 26–81 years). Gender and age were not significantly associated with EoR rate (*p* = 1.763) or hospital length of stay. Mean cranio-caudal diameter of lesion was 46.5 mm (range 41–61 mm). According to Knosp [8] grading scale, 30 (31.2%) were classified as grade 0; 24 (25.0%) grade 1; 19 (19.8%) grade 2; 9 (9.4%) grade 3 and 14 (14.6%) grade 4. Endocrinological screening confirmed the non-functioning status in all patients included in the study. Seventy-eight patients (81.2%) presented with visual field defects, 9 of whom (9.4%) also had visual acuity impairment. Fifty-three patients (55.2%) presented with various degree of anterior pituitary insufficiency: 27 had panhypopituitarism, 14 had combined corticotropic and thyrotropic insufficiency, 6 had isolated corticotropic insufficiency, 3 had isolated thyrotropic insufficiency and 3 isolated gonadotropic insufficiency. Twelve patients (12.5%) had headaches and 3 (3.1%) presented drowsiness associated with hydrocephalus. According to intraoperative assessment of lesion consistency, 50 GPAs (52.1%) were soft, whereas 46 GPAs (47.9%) were firm/fibrous. Thirty-one GPAs (32.3%) were deemed to have significant vascularization due to profuse intraoperative bleeding impairing visualization through the endoscopic lens and requiring more than 20 mL of advanced hemostatic matrix, whereas the remaining 65 GPAs (67.7%) did not. Interestingly, our results are in keeping with the data reported by previous studies [11,12], confirming that 2/3 of GPAs are not highly vascularized. EoR was classified as follows: GTR in 34 cases (35.4%) (Figure 1), NTR in 12 cases (12.5%) (Figure 2), and STR in 50 (52.1%) (Figure 3 and Figure 4). 

On univariate analysis, CS extension, large (>2.5 cm from diafragma sellae) suprasellar and/or intraventrincular extension, and firm/fibrous consistence showed to be significant factors in determining the EoR rate (*p* = 0.034, *p* = 0.041, and *p* = 0.037, respectively), while significant vascularization did not influence it. In multivariate analysis maximum diameter of the GPAs (*p* 0.021) and soft consistency of the lesion showed to be statistically significant factor influencing the EoR and favoring GTR (*p* = 0.021 and *p* = 0.043 respectively). All patients with preoperative visual impairment, except one developing a monocular blindness postoperatively, showed an improvement in both visual field and visual acuity deficits (77 patients, 98.7%). Among 53 patients presenting preoperatively various degree of pituitary insufficiency, 27 (50.9%) showed a recovery of their pituitary function within 6 months from surgery. Regarding surgical complications, although a CSF leak was observed intraoperatively in 16 (16.7%) patients, it persisted postoperatively in 7 patients (7.3%), that underwent serial lumbar punctures (up to three deliquorations) or lumbar drain for 3 to 5 days. In 5 cases rhinorrhea dried up, while in 2 cases it was necessary a re-do surgical exploration, which revealed that the CSF fistula was caused by a necrosis of the pedicled flap used for primary repair, hence requiring rescue with a contralateral pedicled nasoseptal flap. Seven patients (7.3%) presented with postoperative bacterial meningitis, which was associated to postoperative CSF leakage in 3 cases only. All cases were successfully treated by targeted antibiotic therapy. Two patients (2.1%) had apoplexy of the residual tumor on 3rd and 5th postoperative day, respectively. Clinical presentation included abrupt headache, meningeal signs, severe hyperthermia, tachypnea with systemic alcalosis, arterial hypertension, 3rd nerve palsy, and DI. Both patients underwent prompt management in our endocrinology units with high dose steroids and close monitoring of vital parameters. In both cases, clinical improvement occurred within 7 days from symptoms onset. One case of postoperative monocular blindness (1%) and one case of postoperative hydrocephalus requiring ventriculoperitoneal shunt (1%) were recorded. Thirty-six patients (37.5%) developed transient diabetes insipidus, which resolved in 4 to 12 days after surgery, while delayed postoperative hyponatremia was observed in 20 patients (20.8%) and was resolved in all cases within 1 week from diagnosis. At a mean follow-up of 52.4 months (range: 24–88 months), 16 patients (16.7%) with a known residual tumor showed a progression of the disease after a mean recurrence-free survival of 32 months. Re-do EEA was carried on in 3 cases (3.1%), whereas 13 (13.5%) were referred for adjuvant radiotherapy (SRT for 9 patients, 9.4% and Gamma Knife SRS for 4 patients, 4.2%).

## 4. Discussion

GPAs are relatively uncommon, representing 5–14% of all adenomas [6,18,19], and accounted for 8.1% of cases in our multicenter series. Because of their extension beyond sella turcica and tendency to encase and infiltrate relevant nervous and vascular structures, GPAs constitute a significant diagnostic and therapeutic challenge [20,21]. Tumoral resection, although technically challenging, constitutes the mainstay for treatment and is usually performed in tertiary centers with large volumes of skull base referrals and SRS or SRT facilities. Since complete resection can be achieved in only a minority of patients, and NTR and STR are reported in less than 50% in most series, a multimodality approach is usually considered, with fractionated stereotactic radiosurgery or stereotactic radiotherapy as adjuvant treatment in most cases of both secreting and non-secreting GPAs [22,23,24,25]. Even though the transcranial routes via pterional, subfrontal, or fronto-orbitozygomatic approaches represent still a valid alternative to remove the parasellar tumoral component, they provide only limited exposure of the intrasellar region. In addition, those procedures carry the burden of a higher complication rate when it comes to a comparison with the transsphenoidal route [20,21,26]. Transsphenoidal approaches have been widely implemented and are nowadays considered safe and effective approaches to the intra- and parasellar regions [26,27,28,29]. Although the transsphenoidal microscope-assisted approach was the first to be developed and widely used in the past [20,21], most recently, the use of EEA-TPP to reach the sellar and parasellar spaces and the development of enhanced endoscopic technology with high-definition cameras, screens, and endoscopes lighting [17,18,30,31] have improved quality performance metrics of endoscopic surgery [32]. However, supra- and lateral sellar extension, fibrous texture, and significant vascularization make endoscopic excision difficult, potentially dangerous, and seem to be correlated with incomplete EoR [31]. In such cases, a two-staged transsphenoidal approach has been proposed, entailing initial debulking of the tumor followed by tackling the further descent of the upper part of the GPA secondarily [32]. However, such a strategy carries the disadvantage of repeated surgeries, potential risk for swelling, and early postoperative bleeding of residual tumor, with acute hydrocephalus and increased optic nerve compression. In recent literature several reports of EEA treatment of GPAs are available [33,34]. Koutourousiou et al. [35] in a retrospective study on 54 GPAs endoscopically treated achieved globally a 20.4% rate of GTR and a 66.7% of STR respectively. Yano et al. [36] reported the results on a 34 patients’ series of GPAs obtaining a STR rate of 47.1%. Elshazly et al. [37], analyzing a series of 55 patients, reported a GTR rate of 44%, a STR of 47% and a partial removal of 9%, Nakao et al. [38] were able to get a GTR rate of 47% and a STR of 53% on a series of 43 patients, while Kuo et al. [39] reported a GTR rate of 20.5% among 38 patients. In line with previous reports, the present study recorded a GTR or NTR in nearly 48% of cases. Several authors highlighted that a multilobular or irregular tumor configuration and cavernous sinus extension were important factors limiting the quality of tumor resection, while tumor size, extension to the ventricular system and the anterior or posterior fossa, were not [35,36,37]. We confirmed that significant factors determining EoR were CS and large suprasellar extensions, demonstrating also that soft consistency of the lesion is a significant factor for GTR. However, the objective of maximal tumor resection for GPAs is outranked by the scope of relieving mass effect on cranial nerves, CSF pathways, frontal lobes, hypothalamus, and pituitary gland to achieve clinical improvement and avoid surgical complications. In fact, also our low rate of perioperative CSF leak, the most common complication encountered in EEA, occurring on average in 16.7% of the cases [19,35,36,37,38,39] could be attributed to our surgical strategy, which consisted of using pedicled nasoseptal flap and only rarely extracapsular resection, favoring safety over radical resection, and considering patients for radiation treatment in case of residual tumor. In fact, although some authors consider subtotal resection a protective factor from perioperative CSF leak, the residual tumor acts as a plug in the subarachnoid spaces that renders pedicled flap for reconstruction unnecessary [40]. 

Postoperative visual field deficit improvement rate varies in the literature, and it is more likely to be obtained with transsphenoidal than transcranial surgery [21,41]. In a review by Marigil Sanchez et al. [42] analyzing 12 studies reporting purely endoscopic management of GPAs, visual improvement ranged between 71% and 92%. In the present series, most patients presented with visual field and visual acuity disturbances, nonetheless all but one of them demonstrated improvement after EEA-TTP regardless of EoR. Even the patient who suffered from postoperative monocular right eye blindness, most probably due to ischemic optic nerve injury, experienced an improvement on the contralateral eye vision compared to baseline. In literature, postoperative visual worsening after endoscopic transsphenoidal surgery varies between 2% and 3.7% [35,43], while it increases up to 22% after transcranial GPAs resection, probably due to optic apparatus manipulation [21,44,45] (Table 2).

Koutourousiou et al. [35] found that all visual worsening in their series of endoscopically resected GPAs were caused by residual tumor apoplexy. Our 2 patients with residual tumor apoplexy did not show visual impairment but transient meningism, 3rd cranial nerve palsy, and signs of hypothalamic/pituitary stalk compression, not requiring reoperation, as bleeding was mainly contained within the CS and evolved favorably. However, apoplexy of residual adenoma, ranging between 3.2% and 18.7%, is a dreadful complication, and represents the most reported cause of postoperative mortality in GPAs patients [21,47]. This condition should be managed with surgical decompression if supportive therapy with intravenous fluids and corticosteroids is ineffective [46]. High level of suspicion should be maintained in case of partial resection of GPAs in the first 48–72 h postoperatively and efforts should be made intraoperatively to reduce the amount of residual tumor in the subarachnoid spaces. Although it is appropriate to rapidly mobilize all GPAs patients in the immediate postoperative period and start thromboprophylaxis to tackle the increased thromboembolic risk due to sudden decrease in cortisol levels, responsible for a pro-inflammatory, pro-thrombotic state, attention should be paid to weight the benefits of pharmacological thromboprophylaxis in patients at higher risk for apoplexy. The key for successful management of this potentially life-threatening complication is the effective cooperation between neurosurgical and neuroendocrinological teams, leading in most severe cases to admission to dedicated endocrinological high dependency or intensive care unit for close monitoring and aggressive medical treatment. Even though anterior pituitary insufficiency is a common presenting condition in case of GPAs, accounting for more than 50% of cases in our and other authors’ series [20,48,49,50,51] and transient and permanent hypopituitarism are well known complications of EEA, their rate being similar to microscopic transsphenoidal approach, but lower compared to transcranial surgery [21,26,41], postoperative endocrinological outcome for GPAs has not been explored in detail. Most of the published clinical studies report heterogeneous cohorts of patients, including functioning and non-functioning GPAs [35,36,37,38,39,42]. Our series included only non-functioning GPAs, although functioning status was not considered among criteria for exclusion. This allowed a less biased analysis of surgical outcome, as none of our patients underwent pre- or postoperative long-term medical treatment. In the present series, a postoperative pituitary function improvement was recorded in the majority (50.9%) of patients, especially for preoperative adrenal insufficiency. Hormonal improvement in macroadenomas ranges between 35% and 50%, an estimation that may not be applicable to giant lesions, in which hypopituitarism is generally long-standing and therefore more difficult to recover [46,47]. 

## 5. Conclusions

Management of GPAs is challenging and can be hardly standardized, as many factors affect surgical outcome. Therefore, a multimodal strategy should be tailored on a case-by-case basis. Although long-term disease control often needs adjuvant treatment, the initial course of action consists in an aggressive maximal resection, which should be pursued, with the aim of relieving mass effect in order to improve visual and endocrinologic function. EEA-TTP allows satisfactory resection rates, even for lesions extending into the CS, in the suprasellar compartment, in the ventricular and clival regions. However, tumor extension beyond the lateral walls of CS together with a firm/fibrous consistency continues to have negative predictive factors for total/near-total removal. Except for postoperative CSF leak, whose rate may be contained by careful application of pedicled flap reconstruction techniques, incidence of EEA’s complications appears to be lower than those reported from microscopic transnasal and open surgery. Thus, surgery through EEA-TTP approach should be promoted as the initial treatment for GPAs.

## Figures and Tables

**Figure 1 cancers-13-03603-f001:**
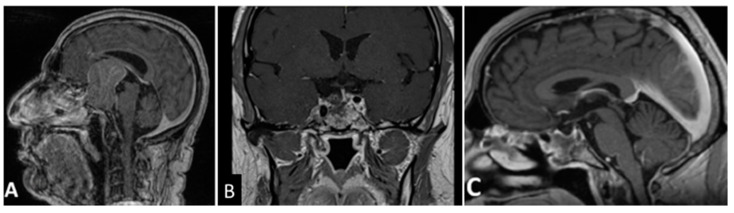
T1- weighted contrast-enhanced MR images. Preoperative sagittal (**A**) view showing a giant pituitary macroadenoma, Knosp grade 1. Three-month postoperative coronal (**B**) and sagittal (**C**) views showing GTR. GTR: gross total removal.

**Figure 2 cancers-13-03603-f002:**
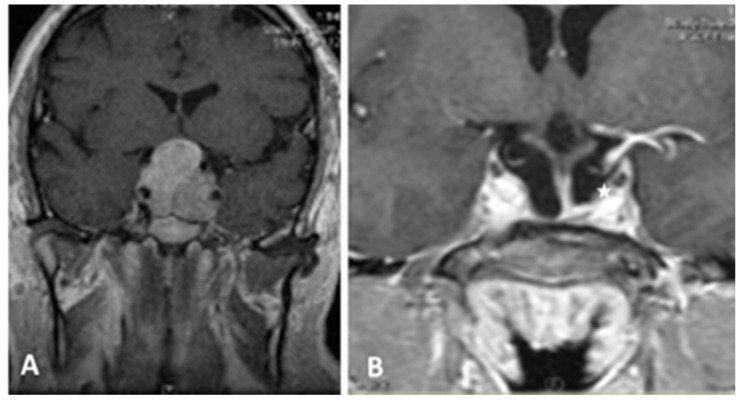
T1- weighted contrast-enhanced coronal MR images. Preoperative view (**A**) showing a giant pituitary macroadenoma, Knosp grade 3. (**B**) Three-month postoperative view showing NTR with a small tumor residue in the medial left cavernous sinus (white asterisk).

**Figure 3 cancers-13-03603-f003:**
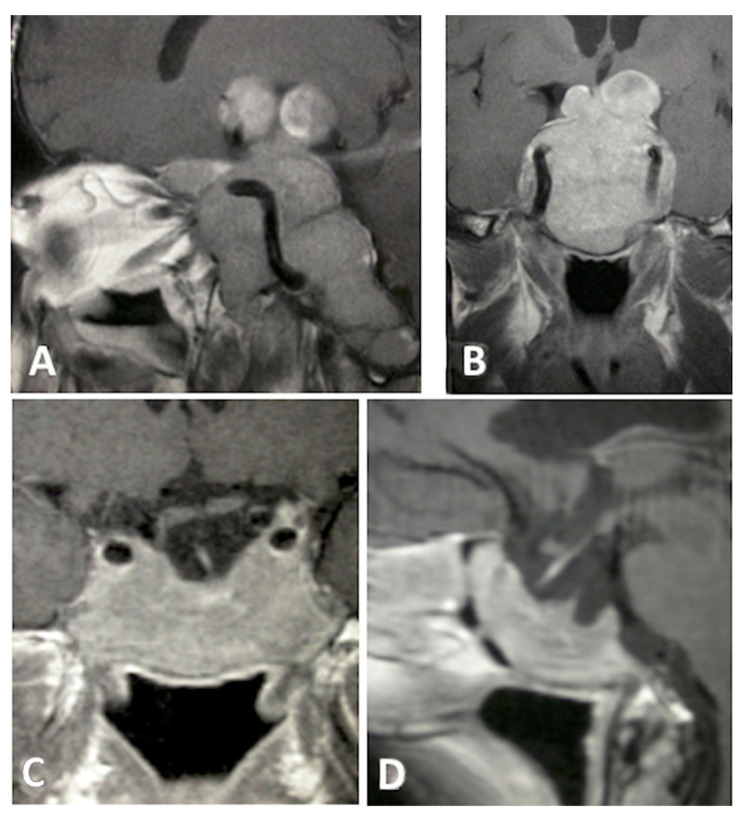
T1-weighted contrast-enhanced MR images. Preoperative sagittal (**A**) and coronal (**B**), views showing a giant pituitary macroadenoma, Knosp grade 4. Three-month postoperative coronal (**C**) and sagittal (**D**) views showing STR with residual tumor in both cavernous sinuses.

**Figure 4 cancers-13-03603-f004:**
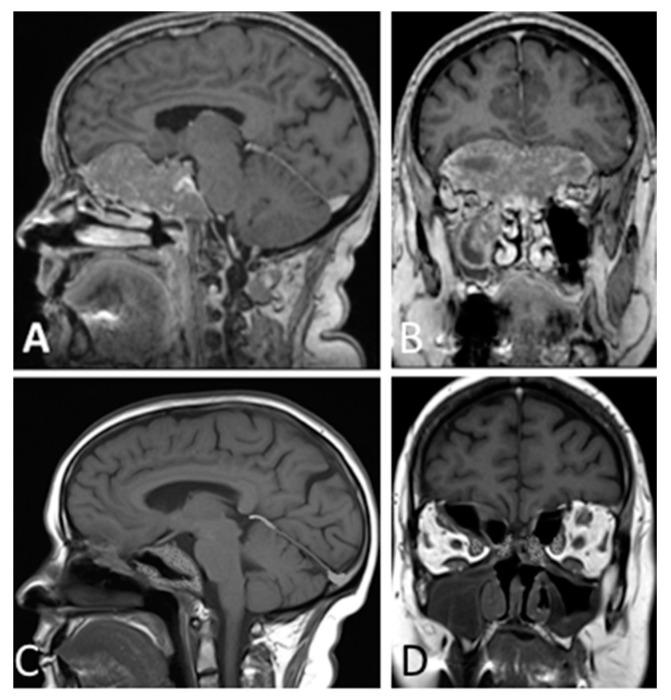
T1- weighted contrast-enhanced MR images. Preoperative sagittal (**A**) and coronal (**B**) views showing a giant pituitary macroadenoma, Knosp grade 3, extended to the right maxillary sinus, subfrontal, and clival areas. Three-month postoperative sagittal (**C**) and coronal (**D**) views showing STR with residual tumor left in the clival, right subfrontal, and right orbital areas.

**Table 1 cancers-13-03603-t001:** Characteristics of a series of 96 giant pituitary adenomas.

Demographics	N.	%
Male	55	57.3
Female	41	42.7
Mean Age (years)	52.2 (26–81)	
Mean tumor diameter (mm)	46.5 (41–61)	
**KNOSP Classification**		
Grade 0	30	31.2
Grade 1	24	25.0
Grade 2	19	19.8
Grade 3	9	9.4
Grade 4	14	14.6
**Endocrinological status**		
Non-functioning	96	100
**Preoperative Clinical symptoms and signs**		
Visual field defects	78	81.2
Anterior pituitary insufficiency	53	55.2
Headache	12	12.5
Visual acuity deficit	9	9.4
Drowsiness	3	3.1
**Treatment**		
EEA-TTP ^1^	96	100
**GPA consistency**		
Soft	50	52.1
Firm/fibrous	46	47.9
**GPA vascularization**		
Significant	31	32.3
Not significant	65	67.7
**Extent of resection**		
GTR ^2^ (100%)	34	35.4
NTR ^3^ (96 to 99%)	12	12.5
STR ^4^ (75 to 95%)	50	52.1
**Clinical outcome**		
Visual improvement	77	98.7
Recovery of pituitary function	27	50.9
**Surgical complications**		
CSF ^5^ leak	7	7.3
Meningitis	7	7.3
Apoplexy of residual tumor	2	2.1
Hydrocephalus	1	1.0
Right eye Blindness	1	1.0
Transient DI	37	37.5
Transient delayed hyponatremia	20	20.8
**Progression**		
Yes	16	16.7
No	80	83.3
**Treatment of 16 recurrences**		
Re-do EEA-TTP	3	3.1
GKRS ^6^	4	4.2
SRT ^7^	9	9.4

^1^ EEA-TTP, extended endoscopic approach transtuberculum/transplanum. ^2^ GTR, gross total removal. ^3^ NTR, near total removal. ^4^ STR, subtotal removal. ^5^ CSF, cerebrospinal fluid. ^6^ GKRS, gamma-knife radiosurgery. ^7^ SRT, stereotactic radiotherapy.

**Table 2 cancers-13-03603-t002:** Surgical indications, technical limitations, and complications of various approaches to GPAs.

Surgical Approaches		EEA ^1^	Microscopic Transsphenoidal	Transcranial
Indications [20,21,22,23,24]		Medial wall cavernous invasion	Retro-chiasmatic extension of the tumor and expansion into the ventricular system	Temporal lobe invasion
		Tumor extending into planum sphenoidale amenable for transplanum EEA approach		Very large or dumbbellshaped tumors (usually more than 50 mm) extending into the planum sphenoidale, middle fossa or retro-chiasmatic region, especially in case of a shallow sella and/or narrow inter-carotid space
Contraindications [9,10,21,22]	Absolute	Tumor extension laterally to the supra-clinoidal part of the ICA ^2^	N/A ^3^	Cavernous invasion
	Relative	N/A ^3^	Cavernous invasion	N/A ^3^
Complications [19,31,41,42,46]	Visual deterioration	+	++	+++
	Postoperative cranial nerve dysfunction	+	++	+++
	Pituitary function amelioration	++	+	+
	Diabetes Insipidus	+	++	+++
	CSF leak	+++	++	+
	Meningitis	+	++	++
	Mortality	+	+	+

^1.^ EEA, extended endoscopic approach. ^2^ ICA, internal carotid artery. ^3^ N/A, not available.

## Data Availability

The statistical analysis plan, study protocol, deidentified data and informed consent of the included patients are available upon motivated request to the corresponding author (ORCID identifier orcid.org/0000-0002-5040-1916) and may be reused to reproduce research, to make public assets available to the public, to leverage investments in research, and to advance research and innovation.

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
