# Peer review of "Primary Endoscopic Endonasal Management of Giant Pituitary Adenomas: Outcome and Pitfalls from a Large Prospective Multicenter Experience"

_cancers, 2021, doi:10.3390/cancers13143603_

Round 1

Reviewer 1 Report

The authors of this manuscript have to be commended for this prospective multicenter study analyzing the outcome after surgery for giant pituitary adenomas (GPA), a challenging condition, via extended endoscopic endonasal transtuberculum/transplanum approach (EEA-TTP). The reviewer would like to suggest some minor improvements prior to the publication of this manuscript:

Figures, general: Consider to omit the axial images.

Figure 2: There is no asterisk.

Tables, general: The indentations appear somewhat arbitrary, rendering the content of the tables difficult to understand.

Knosp Grading: Adding the appropriate reference (DOI: 10.1227/00006123-199310000-00008) is recommended.

Some typos deserve eradication.

Author Response

Reviewer 1
We thank Reviewer 1 for the very positive feedback and appreciated all suggestions provided. As requested: 1) axial images have been taken off from Fig 3; 2) white asterisk has been added to Fig 2; 3) an English native speaker was hired to take care of removing all typos and rephrasing sentences (all changes made are highlighted in red); 4) the reference regarding the Knosp classification [Ref 8] was added.

Reviewer 2 Report

Chibbaro and colleagues have presented data from 5 neurosurgical centers that convincingly show that the extended endoscopic endonasal transtuberculum/transplanum approach should be promoted as primary treatment for giant pituitary adenomas (GPAs). The study is well done and highly informative for neurosurgeans dealing with these tumor entities.

Minor comment

Due to their size, GPAs are very well vascularized tumors. In the paper, it is mentioned that tumors were classified as significantly and non-significantly vascularized. What have been the criteria for this classification? 

Author Response

Reviewer 2

We thank Reviewer 2 for providing very good comments on our manuscript. We have rephrased the sentence regarding vascularization of GPAs in the Materials and Methods and expanded on this aspect in the Result section. It is now clearer that vascularization was also assessed intraoperatively and on video recordings as significant or non-significant depending on degree of intraoperative bleeding appreciated through the endoscopic lens. Our criteria for considering a given GPA highly vascularized were continuous impairment of the visualization through the endoscopic lens and use of more than 20ml of advanced hemostatic matrix. Contrary to Rutland et al. [Ref 11] who evaluated the degree of vascularization on postoperative histological examination, and Cuoccolo et al. [Ref 12] who assessed it on preoperative MRI scans, at time of drafting the study protocol we opted for an intraoperative assessment of vascularization, because this represented an area not well covered by previous studies. Interestingly, our results are in keeping with the data reported by the abovementioned studies with 2/3 of GPAs not being highly vascularized.  

Reviewer 3 Report

cancers-1227128

Chibbaro and colleagues describe a prospective multi-center study on primary endoscopic endonasal management of giant non-functional pituitary adenomas. The relevance of the topic is well-described in the introduction, although the results as presented are impressive as in our experience, in line with literature, morbidity and mortality figures are slightly higher. Moreover, I have the following major comments.

Major comments

  • The authors describe transient DI, but do not mention any permanent DI. Furthermore, the authors describe the improvement of pituitary function, but it is likely that in some patients there must have been some deterioration in pituitary function. Please comment.
  • As all patient underwent EEA-TTP approach with opening of the dura the peri-operative CSF-leakage number seems somewhat low in our experience. I believe the authors should be able to comment on this. To be clear, I don’t comment on the post-op CSF leakage.
  • As it is a multi-center study covering a 5 year period the average amount of surgically treated adenomas per center is 48 patients. As it is important to stress that the message of this paper should not be that this complicated approach should be performed in low volume centers, it might be suitable to mention the contribution of each individual center to the study.
  • As mentioned before these results are outstanding and I congratulate the authors with this. However, I think that the complications of a subtotal resection and possible apoplexia should be more emphasized in the discussion and/or introduction
  • I agree completely with the authors that the EEA-TTP in many cases is the best approach for giant adenomas. As the EEA-TTP is performed in every case in this series, maybe the authors can comment on this in the discussion.

Minor comments;

  • Figure 4 the post-op MRI is without contrast and therefor suboptimal.
  • In table 2 Cavernous invasion is covering 2 cells.

Author Response

X

Reviewer 3

    We thank Reviewer 3 for congratulating us on this study and the excellent food for thoughts provided. 
We took all points raised into account and we have listed all our corrections and answers down here: 

-    With regards to pituitary function deterioration, in the present cohort, only few patients showed a transient deterioration (clinical worsening associated with off range laboratory values) that regressed in all cases within 6 months (of note, regression was declared by our endocrinologists as soon as hydrocortisone and others hormone replacement could be discontinued). Because none of the patients included in this series required steroids or hormone replacement, the discussion about postoperative deterioration of pituitary function is limited to cases who experienced a transient worsening.  
-    In this series we recorded only 16 cases of perioperative CSF leak. Such low figure could be attributed to our standard surgical technique using only rarely the extracapsular resection as well as our GTR of 35.4%. Subtotal resection is known to be a protective factor from perioperative CSF leak: in fact experts do not even suggest flap reconstruction or use of fat/sealant in those scenarios (Ref: Cavallo LM, Solari D, Somma T, Cappabianca P. The 3F (Fat, Flap, and Flash) Technique For Skull Base Reconstruction After Endoscopic Endonasal Suprasellar Approach. World Neurosurg. 2019 Jun;126:439-446. doi: 10.1016/j.wneu.2019.03.125) 
-    We completely agree that GPAs are very complex and challenging lesions, which should be managed in specialized tertiary centers. All neurosurgical departments taking a part in the present prospective clinical study are tertiary neurosurgical centers equipped with a neuroendocrinology unit, a dedicated skull base team (ENT and neurosurgery) and a large referral base of various skull base tumors including pituitary. All units have extensive experience in the comprehensive management of pituitary adenomas (not only from a surgical and endoscopic perspective but also in terms of stereotactic radiotherapy/radiosurgery). To highlight such aspect of GPAs management we have discussed the importance of centralizing complex cases in tertiary centers able to provide a holistic approach to those lesions. 
-    As suggested, the role of postoperative complications and apoplexy have been discussed more in details in the manuscript.
-    As requested, the images and formatting of the Tables have been improved. 

Round 2

Reviewer 3 Report

Dear colleagues,

Thank you for your adequate respond. Although I still would like to discuss some of the results, the given answers are satisfying. congratulations with this nice paper.